# Prediction of Coronary Artery Calcium Score Using Machine Learning in a Healthy Population

**DOI:** 10.3390/jpm10030096

**Published:** 2020-08-20

**Authors:** Jongseok Lee, Jae-Sung Lim, Younggi Chu, Chang Hee Lee, Ohk-Hyun Ryu, Hyun Hee Choi, Yong Soon Park, Chulho Kim

**Affiliations:** 1School of Business Administration, Hallym University, Chuncheon 24252, Korea; ljs1844@hallym.ac.kr (J.L.); 41774@hallym.ac.kr (C.H.L.); 2Department of Neurology, Hallym University Sacred Heart Hospital, Anyang 14068, Korea; jaesunglim@hallym.or.kr; 3Industry-University Cooperation Group, Hallym University, Chuncheon 24252, Korea; 42215@hallym.ac.kr; 4Department of Endocrinology, Chuncheon Sacred Heart Hospital, Chuncheon 24253, Korea; ohryu30@gmail.com; 5Department of Cardiology, Chuncheon Sacred Heart Hospital, Chuncheon 24253, Korea; jumdure@hallym.or.kr; 6Department of Family Medicine, Chuncheon Sacred Heart Hospital, Chuncheon 24253, Korea; pyongs@hallym.ac.kr; 7Department of Neurology, Chuncheon Sacred Heart Hospital, Chuncheon 24253, Korea

**Keywords:** cardiovascular disease, machine learning, coronary artery calcium score, coronary computed tomography angiography

## Abstract

Background: Coronary artery calcium score (CACS) is a reliable predictor for future cardiovascular disease risk. Although deep learning studies using computed tomography (CT) images to predict CACS have been reported, no study has assessed the feasibility of machine learning (ML) algorithms to predict the CACS using clinical variables in a healthy general population. Therefore, we aimed to assess whether ML algorithms other than binary logistic regression (BLR) could predict high CACS in a healthy population with general health examination data. Methods: This retrospective observational study included participants who had regular health screening including coronary CT angiography. High CACS was defined by the Agatston score ≥ 100. Univariable and multivariable BLR was performed to assess predictors for high CACS in the entire dataset. When performing ML prediction for high CACS, the dataset was randomly divided into a training and test dataset with a 7:3 ratio. BLR, catboost, and xgboost algorithms with 5-fold cross-validation and grid search technique were used to find the best performing classifier. Performance comparison of each ML algorithm was evaluated with the area under the receiver operating characteristic (AUROC) curve. Results: A total of 2133 participants were included in the final analysis. Mean age and proportion of male sex were 55.4 ± 11.3 years and 1483 (69.5%), respectively. In multivariable BLR analysis, age (odds ratio [OR], 1.12; 95% confidence interval [CI], 1.10–1.15, *p* < 0.001), male sex (OR, 2.91; 95% CI, 1.57–5.38, *p* < 0.001), systolic blood pressure (OR, 1.02; 95% CI, 1.00–1.03, *p* = 0.019), and low-density lipoprotein cholesterol (OR, 1.00; 95% CI, 0.99–1.00, *p* = 0.047) were significant predictors for high CACS. Performance in predicting high CACS of xgboost was AUROC of 0.823, followed by catboost (0.750) and BLR (0.585). The comparison of AUROC between xgboost and BLR was significant (*p* for AUROC comparison < 0.001). Conclusions: Xgboost ML algorithm was found to be a more reliable predictor of CACS in healthy participants compared to the BLR algorithm. ML algorithms may be useful for predicting CACS with only laboratory data in healthy participants.

## 1. Introduction

Cardiovascular disease (CVD) is one of the leading causes of death worldwide [1]. Inflammation of the vascular smooth muscle cell results in increased calcium deposits that develops into atherosclerotic plaque on the internal wall of the coronary artery. Further, the internal diameter of the vessels deteriorates due to the expansion and rupture of the calcifying plaque, resulting in cardiovascular disease [2]. When evaluating the risk of coronary atherosclerosis, we usually assess an individual’s conventional risk factors including hypertension, diabetes, dyslipidemia, and smoking. In addition, coronary artery calcium score (CACS) in computed tomography (CT) is an important predictor for future CVD development and mortality in the general population [3,4,5]. It has been reported that an assessment of CACS along with the Framingham Risk Score stratification, can be more useful in assessing future CVD development rather than just evaluating the latter [6]. Likewise, identifying CACS in the general population is a useful tool to identify high risk patients for CVD in primary prevention. 

Machine learning (ML) has recently been adopted in a variety of medical problem solving or outcome predictions because it shows greater accuracy compared to conventional statistical methods due to its use of tremendous computational power [7]. Previous studies have focused on deep learning algorithms to predict CACS using chest CT, which has already shown promising results in image deep learning tasks [8,9]. Moreover, there have been a variety of ML algorithms to classify high and low risk for CVD more efficiently than the conventional logistic regression analysis [10,11,12]. The performance of these ML algorithms are gradually improved by several ML techniques such as data scaling/normalization, outlier or noise processing, and cross-validation to minimize overfitting and underfitting [13]. Recently, Al’Aref et al. suggested that an ML model including clinical features and CACS can improve the future coronary artery obstructive disease (CAOD) risks in 35281 Coronary CT Angiography Evaluation for Clinical Outcomes: An International Multicenter (CONFIRM) registry patients [10]. However, to our best knowledge, no study has assessed the feasibility of ML algorithms to predict the CACS using only clinical variables in a healthy general population. Therefore, the aim of this study was to assess the performance of several ML classifiers to predict CACS in addition to conventional binary logistic regression (BLR) analysis. We hypothesized that classification performances of several ML classifiers are superior to those of BLR analysis.

## 2. Materials and Methods

### 2.1. Study Participants

This is a retrospective observational study using hospital-based participants who had regular health screening. In Korea, the entire population has medical insurance, and those aged 40 and above are allowed to have a general health screening every two years, which is provided by the National Health Insurance Service. At the same time, if there have been any abnormal health results through prior screening, or if the health policyholder or business owner wants additional investigations at a personal cost, then examinations such as a brain MRI or coronary CT angiography (CTA) are made available. We included patients who were examined between January 2014 and December 2019 at three tertiary academic centers. The inclusion criteria were those patients (1) who were 40 years or older and (2) who opted for a coronary CTA in addition to regular health examination simultaneously. The exclusion criteria were (1) missing clinical or laboratory information and (2) the patients with repetitive coronary CTA. In other words, if the patients had undergone repetitive coronary CTA at each regular health examination, we only included information of the last health examination and coronary CTA results. Figure 1 shows the flow chart of the inclusion and exclusion strategy used in this study. The study protocol was approved by the Hallym University Hospital Institutional Review Board (No. 2019-05-010-001). Informed consent was waived by the IRB because we only used fully deidentified data.

### 2.2. Data Collection

As described earlier, we included laboratory parameters performed at regular health examinations including complete blood count, liver and renal function tests, lipid profile, atherosclerosis markers, and screening tests for diabetes. In addition, anthropometric measures such as age, sex, blood pressure, and body mass index were included in the input features of ML tasks. Information on clinical characteristics and risk factors for CVD (hypertension, diabetes, dyslipidemia, current smoking) were not included because laboratory parameters such as blood pressure, fasting blood glucose, glycated hemoglobin, or total and low-density lipoprotein (LDL) cholesterol are already included in the prediction model.

### 2.3. Coronary Artery Calcium Score

Three participating centers performed an ECG-triggering cardiac CT scan of Sensation 64 or Somatom Definition Flash (Siemens Medical Solutions, Forchheim, Germany). All CT scanners performed with 64-detector scanning capability, and the exact parameters of the cardiac CT scan were as follows: tube voltage, 120 kVp; window level, 40; window width, 120; slice thickness, 3 mm. CACS was calculated by semi-automated Agatston method [14]. The primary outcome measure was high CACS, defined as a CACS of 100 or more in coronary CTA. 

### 2.4. Machine Learning

We used anthropometric and laboratory variables from the general health examination as input variables. BLR, catboost, and extreme gradient boost (xgboost) ML algorithms were used to classify high and low CACS of the participants. Especially, catboost and xgboost are ensemble tree-based classifiers, which can minimize the issue of overfitting in tree-based classifiers [15,16,17]. We randomly divided the whole dataset into a training and test dataset with a 7:3 ratio. Proportions of high CACS groups were identically distributed in the training and test dataset. Input variables were not scaled or normalized with preprocessing but entered as row values in the ML task. All ML tasks were performed with 5-fold cross-validation, which prevents overestimation of the parameters and reduces information leakage. We used a grid search technique for hyperparameter optimization of the ML algorithms. From the ML training process, we extracted feature importance information of each ML classifier and compared them with each other. 

### 2.5. Statistical Methods

Baseline characteristics of the participants according to high vs low CACS were compared with Student’s *t*-test, Mann–Whitney U, or Pearson’s χ^2^-test, as appropriate. We performed univariable and multivariable BLR analysis for the whole dataset. Statistically significant features with a *p*-value less than 0.05 in the univariable logistic regression analysis were entered into the multivariable model. Predictors in BLR analysis were represented with odds ratio (OR) and 95% confidence interval (CI). 

After the BLR analysis of the whole dataset, the ML classifier was trained in the 4-fold training dataset and validated in the 1-fold dataset, randomly. Performance of each ML classifier was evaluated in unseen test data. We calculated the probability score of the data in each ML algorithm and allocated those with a score of more than 0.5 into the high CACS group. The performance of each ML algorithm was measured by the area under the receiver operating characteristic (AUROC) curve. ML classification tasks were performed with R version 3.6.1 (the R Foundation for Statistical Computing) using moonBook, caret, xgboost, and catboost R packages.

## 3. Results

A total of 2123 participants were included in the final analysis. Mean (±standard deviation) age and proportion of male sex were 55.4 ± 11.3 years and 1483 (69.5%), respectively. Among all participants, 237 participants (11.2%) had a high CACS. Comparison of the anthropometric and laboratory difference for high versus low CACS groups in the whole dataset are shown in Table 1. The high CACS group is likely to be older, shorter in height and male and have a longer abdominal circumference and higher systolic and diastolic blood pressure than those with low CACS. Fasting blood sugar, glycated hemoglobin, lactate dehydrogenase, serum glutamic-oxaloacetic transaminase, blood urea nitrogen, creatinine, and mean corpuscular volume of red blood cells were higher, and total bilirubin, estimated glomerular filtration rate, total cholesterol, LDL cholesterol, and platelet count were lower in the high CACS group than in the low CACS group.

### 3.1. Predictors of High Coronary Artery Calcium Score

Table 2 showed the results of the univariable and multivariable BLR analysis of the whole dataset. Among them, age (OR, 1.12; 95% CI, 1.10–1.15 per 1 years), male sex (OR, 2.93; 95% CI, 1.59–5.40), systolic blood pressure (OR, 1.02; 95% CI, 1.00–1.03 per 1 mmHg), and LDL cholesterol (OR, 1.00; 95% CI, 0.99–1.00) were significant predictors for high CACS.

### 3.2. Performance of Machine Learning Classifier of High Coronary Artery Calcium Score

Appendix A shows the comparison of input variables between the training and test datasets. There were no differences among input features with the exception of triglyceride level. In BLR analysis that included only the training dataset, age and male sex were significant predictors for high CACS (Appendix A). Appendix A depicts the catboost and xgboost ML algorithms detailing the hyperparameters of the best classifier. Figure 2 shows the overall performance comparison of the AUROC of the three algorithms that predict high CACS. The performance of xgboost was better than that of the BLR classifier (*p* for AUROC comparison = 0.008). Figure 3 summarizes the feature importance of each ML model. Age, systolic blood pressure, and LDL cholesterol were significant features in all three ML classifiers. Male sex was found to be a significant predictor of high CACS in BLR analysis but was not a significant feature in xgboost ML classifiers. When we additionally performed ML prediction with the 10 variables that had the highest importance values (Figure 3), the AUROC values did not differ much from the original models (0.605 for binary logistic stress, 0.749 for catboost, and 0.822 for xgboost).

## 4. Discussion

In this study of CACS prediction using health examination data, age, male sex, and systolic blood pressure were significant predictors for high CACS in a healthy population. BLR analysis provides reasonable information for the significant predictors of high CACS, but its prediction performance was lacking as a prediction classifier. However, other ML algorithms such as catboost and xgboost classifiers overcome the prediction power of the BLR classifier and provide a satisfactory binary classification prediction.

There have been well-known risk calculators in stratifying CVD risk in symptomatic and asymptomatic patients [18,19,20]. Diamond and Forrester originally developed the Duke Clinical Score using logistic regression, which included age, sex, type of chest pain, and several risk factors for CVD [21]. After reporting the overestimation of the pretest probability of the Duke Clinical Score in predicting symptomatic CAOD, the CAD Consortium score and the CONFIRM Risk score were developed and still used age, sex, and several risk factors for CVD as predictors to predict high CAD risk in patients with CAOD [22,23,24]. In our ML prediction model, age, sex, and several risk factors also contribute to classification prediction. Among them, age, systolic blood pressure, and LDL cholesterol were important features for all three ML classification predictions, which is concordant with previously developed risk stratification calculators. 

In our BLR of high CACS in all participants, LDL cholesterol had a negative association with high CACS. In our data, the high CACS group had lower total and LDL cholesterol levels (184 mg/dL and 86 mg/dL, respectively) compared to the low CACS group, which was different to other study populations [4,6,25]. This phenomenon could be explained by the possibility that those with high CACS may have been taking more lipid lowering agents for primary prevention of CVD than those with low CACS. Further, our data could be partially explained by the fact that high aspartate transaminase/alanine aminotransferase ratio and increased lactate dehydrogenase in individuals with high CACS have been associated with a higher proportion of statin-induced muscle or liver injury than in those with low CACS [26,27]. If drug history was included as an input feature of our ML algorithms, the performance of these models might have been better than current models. However, since we did not include history of previous prescription including lipid lowering agents, this hypothesis should be verified in future studies. 

In our xgboost algorithms, male sex was not a significant predictor of CAS. Furthermore, why is the importance of input features different in predicting high CACS group for each ML model? Logistic regression is used to explain various phenomena with odds ratio but always takes into account the interaction effect when constructing the model [28]. In contrast, catboost and xgboost do not consider interaction effects because these algorithms are composed of tree-ensemble structures, which minimize the feature interaction in classification or regression tasks [15,29,30]. In addition, identifying the importance of catboost and xgboost features could give us novel feature information for CACS prediction. In our result, alkaline phosphatase, platelet count, estimated glomerular filtration rate, body mass index, and white blood cells were importance features in ML algorithms. Alkaline phosphatase is already one of the risk factors or treatment target for cardiovascular disease in addition to the highly sensitive C-reactive protein as atherosclerotic biomarkers [31]. Likewise, implementation of ML other than logistic regression could be useful to improve prediction performance and to find novel associations between input features and target disease.

Our study has some limitations. We did not extract or use information on drug usage, regular exercise, or compliance of primary prevention of CVD. In other words, the disadvantage of this study is that there is no information on statin usage and regular exercise, which are associated with CVD development. Second, this study was conducted on the Korean people, and therefore, it may not be possible to generalize its results to other ethnic groups. When using our method on data from other ethnic groups, we should keep in mind that the probability of several factors contributing to CVD development in Asians is underestimated. This study was conducted on patients attending tertiary hospitals, and therefore, there is a possibility of selection bias even though the participants were likely to be a healthy general population. 

Despite these limitations, our study has some strengths. First, physically healthy participants were the main population in our study. Most other CVD risk scoring systems have been developed for symptomatic CVD patients. Therefore, in order to screen for the risk of CAOD in asymptomatic patients, there was a need to study patients without CVD, rather than those with CVD. Second, our ML algorithms showed improved performance of some ML classifiers compared to logistic regression even though no clinical variables, except age and sex, were used in the training process. Therefore, our results could be useful in predicting future CVD risk in patients with only limited laboratory data.

## 5. Conclusions

In this study, we found that when predicting CACS with laboratory data in a healthy participant, logistic regression was not enough for classifying them into a high CACS group. Instead, the xgboost algorithm can improve the prediction power in CACS estimation.

## Figures and Tables

**Figure 1 jpm-10-00096-f001:**
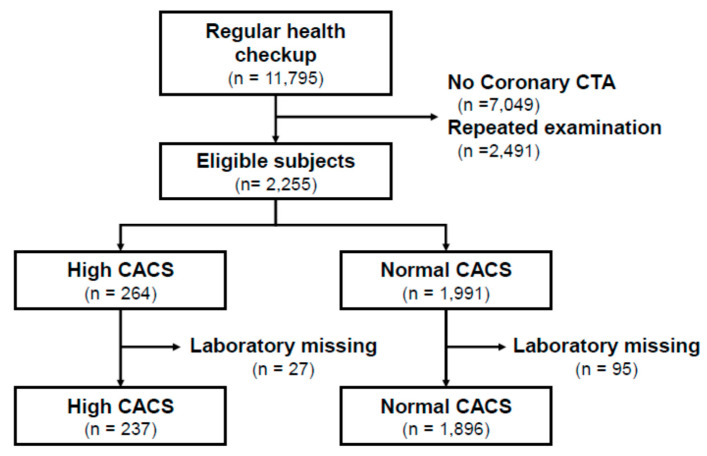
Flow chart of the participants. CTA, Computed tomography angiography; CACS, coronary artery calcium score.

**Figure 2 jpm-10-00096-f002:**
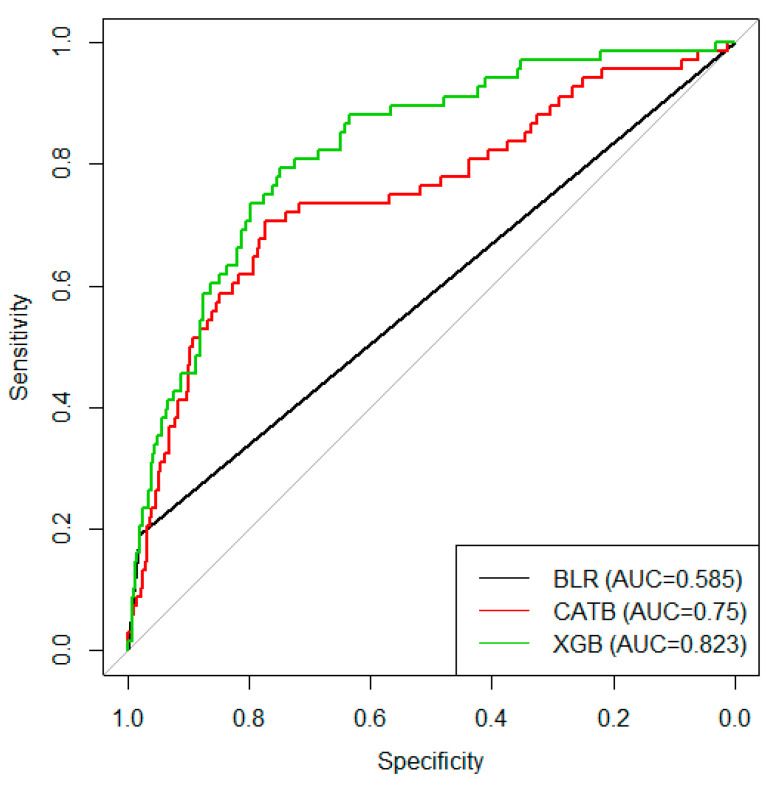
Results of the area under the receiver operating characteristic curves showing the performance of the binary logistic regression, catboost, and xgboost machine algorithms to predict high coronary artery calcium scores. BLR, binary logistic regression; CATB, catboost; XGB, xgboost; AUC, area under the receiver operating characteristic curves.

**Figure 3 jpm-10-00096-f003:**
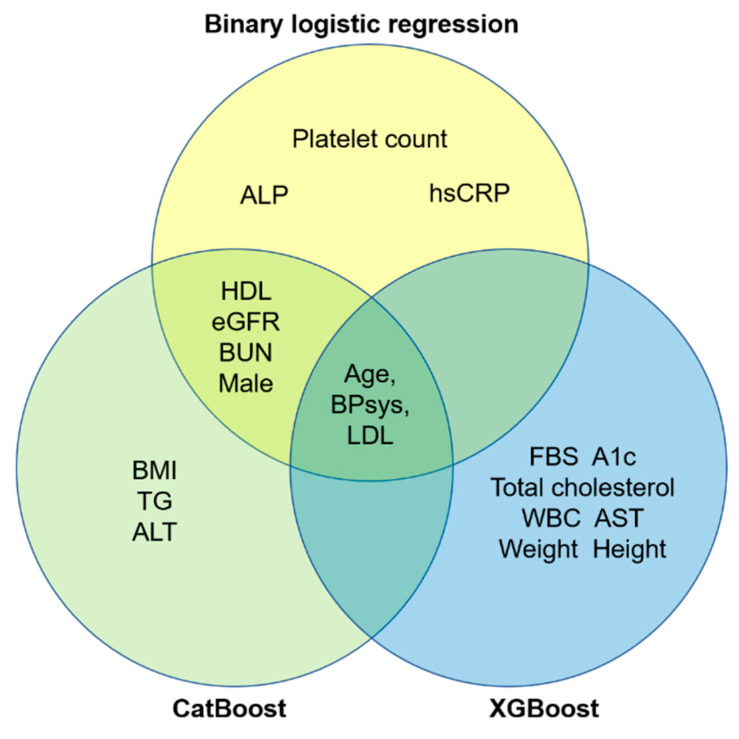
Variable importance plot in each machine algorithm. ALP, alkaline phosphatase; hsCRP, high sensitive C-reactive protein; HDL, high-density lipoprotein; eGFR, estimated glomerular filtration rate; BUN, blood urea nitrogen; BPsys, systolic blood pressure; LDL, low-density lipoprotein; BMI, body mass index; ALT, alanine aminotransferase; FBS, fasting blood sugar; WBC, white blood cell; AST, aspartate transaminase.

**Table 1 jpm-10-00096-t001:** Baseline characteristics of the total study population.

	Low CACS (*n* = 1896)	High CACS (*n* = 237)	Total (*n* = 2133)	*p* Value
Age, years	54.0 ± 10.8	66.2 ± 9.5	55.4 ± 11.3	<0.001
Male	1300 (68.6)	183 (77.2)	1483 (69.5)	0.008
Height, cm	166.3 ± 8.8	164.6 ± 9.0	166.1 ± 8.8	0.007
Weight, Kg	69.2 ± 11.9	68.1 ± 11.0	69.1 ± 11.8	0.187
Abdominal circumference, cm	85.2 ± 9.2	87.0 ± 8.8	85.4 ± 9.2	0.004
BMI, Kg/m^2^	24.9 ± 3.0	25.0 ± 2.9	24.9 ± 3.0	0.514
BP systolic, mmHg	120.8 ± 14.3	128.0 ± 15.5	121.6 ± 14.6	<0.001
BP diastolic, mmHg	73.3 ± 11.2	74.8 ± 10.9	73.4 ± 11.1	0.047
hsCRP, IU/L	1.3 ± 2.4	1.2 ± 2.3	1.3 ± 2.4	0.764
FBS, mg/dL	102.0 ± 24.0	111.4 ± 30.5	103.1 ± 25.0	<0.001
A1c, %	5.6 ± 0.8	5.9 ± 0.9	5.7 ± 0.8	<0.001
Bilirubin (total), mg/dL	0.9 ± 0.3	0.8 ± 0.3	0.9 ± 0.4	0.014
Bilirubin (direct), mg/dL	0.2 ± 0.1	0.2 ± 0.1	0.2 ± 0.1	0.023
gamma-GT, IU/L	44.4 ± 59.3	48.1 ± 66.0	44.8 ± 60.1	0.408
ALP, IU/L	72.4 ± 25.1	73.3 ± 25.4	72.5 ± 25.2	0.572
LDH, IU/L	217.1 ± 83.2	233.7 ± 81.6	219.0 ± 83.2	0.004
AST, IU/L	28.7 ± 29.6	31.1 ± 15.4	28.9 ± 28.4	0.041
ALT, IU/L	31.0 ± 43.0	29.9 ± 22.0	30.9 ± 41.3	0.523
BUN, mg/dL	13.5 ± 3.3	14.8 ± 3.8	13.6 ± 3.3	<0.001
Creatinine, mg/dL	0.9 ± 0.3	0.9 ± 0.2	0.9 ± 0.3	0.015
eGFR, mL/min	85.8 ± 27.3	77.9 ± 25.6	84.9 ± 27.2	<0.001
Total cholesterol, mg/dL	195.5 ± 38.1	184.1 ± 42.5	194.2 ± 38.8	<0.001
TG, mg/dL	145.0 ± 100.3	135.3 ± 86.7	144.0 ± 100.0	0.112
HDL, mg/dL	52.5 ± 13.1	52.1 ± 12.2	52.5 ± 13.0	0.610
LDL, mg/dL	110.4 ± 51.4	85.5 ± 58.1	107.6 ± 52.7	<0.001
WBC, 10^3^/μL	5.7 ± 1.5	5.7 ± 1.6	5.7 ± 1.5	0.918
Hemoglobin, g/dL	14.7 ± 1.4	14.6 ± 1.5	14.7 ± 1.5	0.122
MCV, fL	91.6 ± 4.3	92.5 ± 4.2	91.7 ± 4.3	0.001
Platelet count, 10^3^/μL	243.3 ± 49.4	233.5 ± 44.6	242.2 ± 49.0	0.002
CACS *				<0.001

Values were presented as mean ± standard deviation or number (column percent) as appropriate. * median (interquartile range). CACS, coronary artery calcium score; BMI, body mass index, BP, blood pressure; hsCRP, high sensitivity C-reactive protein; FBS, fasting blood sugar; A1c, glycated hemoglobin; gamma-GT, gamma-glutamyl transferase; ALP, alkaline phosphatase; LDH, Lactate dehydrogenase; AST, Aspartate transaminase; ALT, alanine aminotransferase; BUN, blood urea nitrogen; eGFR, estimated glomerular filtration rate; TG, triglycerides; HDL, high-density lipoprotein; LDL, low-density lipoprotein; WBC, white blood cell; MCV, mean corpuscular volume; IU, international unit.

**Table 2 jpm-10-00096-t002:** Results of the binary logistic regression analysis to predict high coronary artery calcium score in the entire study population (*N* = 2123).

	Univariable Analysis		Multivariable Analysis	
	OR (95%CI)	*p* Value	OR (95%CI)	*p* Value
Age, years	1.12 (1.10–1.13)	<0.001	1.12 (1.10–1.15)	<0.001
Male	1.55 (1.13–2.14)	0.007	2.93 (1.59–5.40)	<0.001
Height, cm	0.98 (0.96–0.99)	0.007	0.95 (0.82–1.10)	0.488
Weight, Kg	0.99 (0.98–1.00)	0.188	1.06 (0.89–1.27)	0.507
Abdominal circumference, cm	1.02 (1.01–1.04)	0.004	1.02 (0.99–1.06)	0.168
BMI, Kg/m^2^	1.01 (0.97–1.06)	0.514	0.80 (0.49–1.30)	0.367
BP systolic, mmHg	1.03 (1.02–1.04)	<0.001	1.02 (1.00–1.03)	0.022
BP diastolic, mmHg	1.01 (1.00–1.02)	0.047	1.01 (0.99–1.03)	0.475
hsCRP, IU/L	0.99 (0.93–1.05)	0.758	0.95 (0.88–1.03)	0.193
FBS, mg/dL	1.01 (1.01–1.02)	<0.001	1.01 (1.00–1.01)	0.070
A1c, %	1.41 (1.23–1.61)	<0.001	1.06 (0.82–1.36)	0.676
Bilirubin (total), mg/dL	0.63 (0.41–0.97)	0.036	0.84 (0.35–2.03)	0.705
Bilirubin (direct), mg/dL	0.24 (0.06–0.99)	0.049	0.68 (0.04–10.92)	0.787
gamma-GT, IU/L	1.00 (1.00–1.00)	0.370	1.00 (1.00–1.00)	0.254
ALP, IU/L	1.00 (1.00–1.01)	0.572	1.00 (0.99–1.00)	0.182
LDH, IU/L	1.00 (1.00–1.00)	0.004	1.00 (1.00–1.00)	0.812
AST, IU/L	1.00 (1.00–1.01)	0.260	1.01 (0.99–1.02)	0.308
ALT, IU/L	1.00 (0.99–1.00)	0.697	0.99 (0.98–1.00)	0.298
BUN, mg/dL	1.12 (1.08–1.16)	<0.001	1.05 (1.00–1.10)	0.067
Creatinine, mg/dL	1.39 (0.93–2.09)	0.105	1.17 (0.71–1.91)	0.538
eGFR, mL/min	0.99 (0.99–0.99)	<0.001	1.00 (1.00–1.01)	0.268
Total cholesterol, mg/dL	0.99 (0.99–1.00)	<0.001	1.00 (1.00–1.01)	0.882
TG, mg/dL	1.00 (1.00–1.00)	0.155	1.00 (1.00–1.00)	0.438
HDL, mg/dL	1.00 (0.99–1.01)	0.676	0.99 (0.97–1.00)	0.094
LDL, mg/dL	0.00 (0.99–0.99)	<0.001	1.00 (0.99–1.00)	0.047
WBC, 10^3^/μL	1.00 (0.92–1.10)	0.918	1.04 (0.94–1.15)	0.444
Hemoglobin, g/dL	0.93 (0.85–1.02)	0.122	0.94 (0.81–1.10)	0.446
MCV, fL	1.06 (1.02–1.09)	0.001	1.01 (0.97–1.05)	0.627
Platelet count, 10^3^/μL	1.00 (0.99–1.00)	0.004	1.00 (1.00–1.00)	0.478

OR, odds ratio; CI, confidence interval; BMI, body mass index, BP, blood pressure; hsCRP, high sensitivity C-reactive protein; FBS, fasting blood sugar; A1c, glycated hemoglobin; gamma-GT, gamma-glutamyl transferase; ALP, alkaline phosphatase; LDH, Lactate dehydrogenase; AST, Aspartate transaminase; ALT, alanine aminotransferase; BUN, blood urea nitrogen; eGFR, estimated glomerular filtration rate; TG, triglycerides; HDL, high-density lipoprotein; LDL, low-density lipoprotein; WBC, white blood cell; MCV, mean corpuscular volume.

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
