# Peer review of "Prediction of Coronary Artery Calcium Score Using Machine Learning in a Healthy Population"

_jpm, 2020, doi:10.3390/jpm10030096_

Round 1
Reviewer 1 Report
The manuscript by Jongseok Lee et al. is describing the possibilities of machine learning (ML) algorithms to predict CACS in healthy subjects. The way of presentation is satisfactory even though there are some points that need to be addressed before final publication.
1# The very first line in Abstract needs to be edited. Furthermore, authors need to add some background under ‘Background’ sub-heading. There are two sentences but it did not represent background of this research.
2# The authors need to add significance of their result in human welfare under ‘Conclusion’ sub-heading in the Abstract section.
3# I think the author needs to describe the role of coronary artery calcium in healthy individual and how come they develop disease.
4# It is very difficult to understand how the author confirmed the relationship between higher CASC and future heart disease? Among the participants 237 had a higher CASC. From the 237 count how many had developed a heart disease? Are they counted some well-known parameters for assuming a future heart disease condition? If so, then what are those parameters? Author needs to prepare a flow diagram (Like Figure 1) for better presentation of their work.
5# There are several typos. Furthermore, some sentences are not in organised manner. Author should check their manuscript carefully. A premium English correction is suggested.
Author Response
The manuscript by Jongseok Lee et al. is describing the possibilities of machine learning (ML) algorithms to predict CACS in healthy subjects. The way of presentation is satisfactory even though there are some points that need to be addressed before final publication.
1# The very first line in Abstract needs to be edited. Furthermore, authors need to add some background under ‘Background’ sub-heading. There are two sentences but it did not represent background of this research.
Response: Thank you for the comment. As you have noted, there was repeated words (cardiovascular disease) in the first line of the abstract, and the study background was not adequately framed. Therefore, we amended the “Background” section as indicated below:
“Background: Coronary artery calcium score (CACS) is a reliable predictor for future cardiovascular disease risk. Although deep learning studies using computed tomography (CT) images to predict CACS have been reported, no study has assessed the feasibility of machine learning (ML) algorithms to predict the CACS using clinical variables in a general healthy population. Therefore, we aimed to assess whether ML algorithms other than binary logistic regression (BLR) could predict high CACS in a healthy population with a general health examination.”
2# The authors need to add significance of their result in human welfare under ‘Conclusion’ sub-heading in the Abstract section.
Response: Thank you for the comment. Per your suggestion, we added a sentence to highlight the significance of our study. The changes made are indicated below:
“Conclusions: Xgboost ML algorithm was found to be a more reliable predictor of CACS in healthy participants compared to the BLR algorithm. ML algorithms may be useful for predicting CACS with only laboratory data in healthy participants.”
3# I think the author needs to describe the role of coronary artery calcium in healthy individual and how come they develop disease.
Response: Thank you for the comment. We added some sentences to describe how coronary artery calcium develops into coronary disease in the introduction section. The changed paragraph is presented below.
“Cardiovascular disease (CVD) is one of the leading causes of death worldwide [1]. Inflammation of the vascular smooth muscle cell results in increased calcium deposits that develops into atherosclerotic plaque on the internal wall of the coronary artery. Further, the internal diameter of the vessels deteriorates due to the expansion and rupture of the calcifying plaque, resulting in cardiovascular disease [2]. When evaluating the risk of coronary atherosclerosis, we usually assess an individual’s conventional risk factors including hypertension, diabetes, dyslipidemia, and smoking. In addition, coronary artery calcium score (CACS) in computed tomography (CT) is an important predictor for future CVD development and mortality in the general population [3–5]. It has been reported that an assessment of CACS along with the Framingham Risk Score stratification, can be more useful in assessing future CVD development rather than just evaluating the latter [6]. Likewise, identifying CACS in the general population is a useful tool to identify high risk patients for CVD in primary prevention.”
4# It is very difficult to understand how the author confirmed the relationship between higher CASC and future heart disease? Among the participants 237 had a higher CASC. From the 237 count how many had developed a heart disease? Are they counted some well-known parameters for assuming a future heart disease condition? If so, then what are those parameters? Author needs to prepare a flow diagram (Like Figure 1) for better presentation of their work.
Response: Thank you for the comment and I apologize for the confusion. This study aims to predict CACS, the biomarker of CAC, not whether heart disease occurred in the target patient. In fact, it is described in the introduction section that CACS has been established as an important predictor for future CVD development and mortality in the general population.
5# There are several typos. Furthermore, some sentences are not in organised manner. Author should check their manuscript carefully. A premium English correction is suggested.
Response: Thank you for the comment. In the revised manuscript, we have used an English Editing Service to check and correct typos and improve the logical flow of sentences, where necessary.
Reviewer 2 Report
This manuscript presents an interesting work focused on the prediction of coronary artery calcium score (CACS) using selected machine learning algorithms in a sample of a healthy Korean population. The methodology used is clearly described and satisfactorily compared and discussed in the context of other related works. The experimental results are interesting, showing the superiority of ensemble models of xboost and catboost over traditionally used binary logistic regression. Nevertheless, I have a couple of notes and suggestions for paper improvement.
- In the first sentence of the Background, "cardiovascular disease" is repeated twice.
- In the first sentence of section 2 authors unproperly use a conditional manner ("may be", "should"). The present is more appropriate.
- Explanation of decision to exclude clinical characteristics, risk factors for CVD with the aim to use numerical data only (in subsection 2.2) does not make much sense. Nominal attributes can be easily transformed into numerical-friendly by binarisation.
- In Table 2 authors present multivariable analysis for all available features. I suggest using Lasso regularisation for the selection of the most relevant subset of features.
- Authors present in subsection 3.2 and further in discussion significant features identified by particular machine learning algorithms. How do you decide which features are relevant for the ensemble models xboost nad catboost?
- More suitable words need to be used in expressions connected with performance: "Performance ... was more significant" on page 7 and "performance was not competent" (on page 8). Performance can be better or worse, or it can be e.g. satisfactory.
Author Response
This manuscript presents an interesting work focused on the prediction of coronary artery calcium score (CACS) using selected machine learning algorithms in a sample of a healthy Korean population. The methodology used is clearly described and satisfactorily compared and discussed in the context of other related works. The experimental results are interesting, showing the superiority of ensemble models of xgboost and catboost over traditionally used binary logistic regression. Nevertheless, I have a couple of notes and suggestions for paper improvement.
- In the first sentence of the Background, "cardiovascular disease" is repeated twice.
Response: Thank you for bringing this to our attention. We have corrected this typo.
- In the first sentence of section 2 authors unproperly use a conditional manner ("may be", "should"). The present is more appropriate.
Response: Thank you for the comment. The sentence has been corrected.
- Explanation of decision to exclude clinical characteristics, risk factors for CVD with the aim to use numerical data only (in subsection 2.2) does not make much sense. Nominal attributes can be easily transformed into numerical-friendly by binarisation.
Response: Thank you for the comment. The original sentence was confusing. As requested, we modified the sentence to explain the decision to exclude clinical characteristics and risk factors for CVD. The modified sentence is presented below.
Information on clinical characteristics and risk factors for CVD (hypertension, diabetes, dyslipidemia, current smoking) were not included because laboratory parameters such as blood pressure, fasting blood glucose, glycated hemoglobin, or total and low-density lipoprotein (LDL) cholesterol, are already included in the prediction model.
- In Table 2 authors present multivariable analysis for all available features. I suggest using Lasso regularisation for the selection of the most relevant subset of features.
Response: Thank you for the comment. We obtained a variable importance value in all three models and selected only 10 important variables that are presented in Figure 3. The results of predicting high CACS with these variables alone was not significantly different from the original model. A sentence was added to the last part of the results section for clarification.
“When we additionally performed ML prediction with the 10 variables that had the highest importance values (Figure 3), the AUROC values did not differ much from the original models (0.605 for binary logistic stress, 0.749 for catboost, and 0.822 for xgboost).”
- Authors present in subsection 3.2 and further in discussion significant features identified by particular machine learning algorithms. How do you decide which features are relevant for the ensemble models xgboost and catboost?
Response: Thank you for the comment. The R packages have a method (or script) for checking variable importance in each model (e.g., variable importance plot in each algorithm). The highest importance variables are located near the root node of the ensemble algorithm such as catboost or xgboost. The reviewer recommended using LASSO regularization for the selection of features in the previous comment. Per our response to the previous comment, we have added a description outlining the result of the additional ML prediction using only 10 important features (with high variable importance) in each model.
- More suitable words need to be used in expressions connected with performance: "Performance ... was more significant" on page 7 and "performance was not competent" (on page 8). Performance can be better or worse, or it can be e.g. satisfactory.
Response: Thank you for the comment. We have modified the expression related to performance and provided more suitable words per the reviewer’s recommendation.
Finally, we corrected the misrepresented part through the English Editing Service.